# Explainable Diagnosis of Melanoma Based on Localization of Clinical Indicators and Self-Supervised Learning

## Abstract

Melanoma is a prevalent lethal type of cancer that is treatable if diagnosed at early stages of development. Skin lesions are a typical warning signs for diagnosing melanoma at early stage, but they often led to delayed diagnosis due to high similarities of cancerous and benign lesions at early stages of melanoma. Deep learning (DL) has been used to classify skin lesion pictures with a high classification accuracy, but clinical adoption of DL for this task has been quite limited. A major reason is that the decision processes of DL models are often uninterpretable which makes them black boxes that are challenging to trust. We develop an explainable DL architecture for melanoma diagnosis. Our architecture segments input images and generates clinically interpretable melanoma indicator masks that are then used for classification. Since our architecture is trained to mimic expert dermatologists, it generates explainable decisions. We also benefit from self-supervised learning to address the challenge of data annotations which is often expensive and time-consuming in medical domains. Our experiments demonstrate that the proposed architectures matches clinical explanations considerably better than existing architectures and at the same time maintains high classification accuracies.

## 1 Introduction

Melanoma is a prevalent type of skin cancer that can be highly deadly in advanced stages Hodi et al. (2010). For this reason, early detection of melanoma is the most important factor for successful treatment of patients because the 5-year survival rate can reduce from a virtually 100% rate to only 5-19% Sandru et al. (2014) as melanoma progresses. New skin moles or changes in existing moles are the most distinct symptoms of melanoma. However, due to similarity of benign and cancerous moles, melanoma diagnosis is a sensitive task that can be preformed by trained dermatologist. Upon inspecting suspicious moles, dermatologists use dermoscopy and biopsy to examine the pattern of skin lesions for accurate diagnosis. If skin moles are not screened and graded on time, melanoma maybe diagnosed too late. Unfortunately, this situation affects the low-income populations and people of color more significantly due to having more limited access to healthcare even if changes in skin lesions are visually noticed. Advances in deep learning (DL) along with accessibility of smartphones have led to emergence of automatic diagnosis of melanoma using skin lesion photographs acquired by the user Codella et al. (2017); Sultana & Puhan (2018); Li & Shen (2018); Adegun & Viriri (2019); Kassani & Kassani; Naeem et al. (2020); Jojoa Acosta et al. (2021). The results of using DL for melanoma diagnosis are promising when evaluated only in terms of diagnosis accuracy with rates close to those of expert dermatologists. Despite this success, however, adoption of these models in clinical setting has been limited.

A primary challenge for adopting deep learning in clinical tasks is the challenge of interpretability. Deep neural networks (DNNs) sometimes are called "black boxes" because their internal decision-making process is opaque. As a result, it is challenging to convince clinicians to rely on models that are not well-understood. Existing explainability methods Shrikumar et al. (2017); Selvaraju et al. (2017); Zhang et al. (2018); Pope et al. (2019) try to clarify decisions of these black boxes to help users or developers understand the most important areas of the image that the model attends to in making the classification. As shown in Figure 1, an attention map can be visualized in the form of a heatmap, where the importance of each spatial pixel is visualized by its intensity. However, an attention map is not particularly helpful in clinical settings, e.g., Grad-Cam Selvaraju et al. (2017) simply highlights the entire mole in the melanoma image in Figure 1. In other words, the highlighted regions are often too large to show the shape of a clinically interpretable region or indicator, or extremely deviate from the regions of interest to dermatologists.

A reason behind this deficiency is that many explainability methods primarily consider the last DNN layer for heatmap generation, whereas some interpretable features maybe encoded at earlier layers of a trained DNN. More importantly, there is no guarantee that a trained DNN uses human interpretable indicators for decision-making Feather et al. (2019); Avramidis et al. (2022), irrespective of improving DL explainability algorithms. In other words, a model may learn to rely on dramatically different features from those used by humans when trained using high-level labels, yet have high classification accuracy rates.

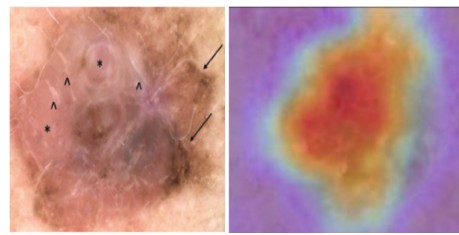

Figure 1: (Left) a skin mole photograph; (right) explainability heatmap generated via Grad-Cam Selvaraju et al. (2017) for a DNN trained to distinguish benign and cancerous moles.

We argue existing explainability methods may not be sufficient for explainable DL in high-stakes domain such as medicine due to the end-to-end training pipeline of DL. In other words, a model that is trained only using a high-level abstract labels, e.g., cancerous vs benign, may learn to extract indicator features that are totally different compared to the features human experts use for classification, i.e., diagnosis. In contrast, dermatologists are trained to perform their diagnosis through identifying intermediate indicator biomarkers that are discovered over decades of accumulative hypothesis testing Argenziano et al. (1998). The solution that we propose is to benefit from intermediate-level annotations that denote human-interpretable features in the training pipeline to enforce a DNN learn a decision-making process more similar to clinical experts. The challenge that we face is that data annotation, particularly in medical applications, is an expensive and time-consuming task and generating a finely annotated dataset is infeasible. To circumvent this challenge, we use self-supervised learning Chen et al. (2020) to train a human-interpretable model using only a small fine-annotated *Dataset A*nd a large coarse-annotated dataset. Our empirical experiments demonstrate that our approach leads to high classification accuracy and can generate explanations more similar to expert dermatologists.

## 2 Related Work

**Explainability in Deep Learning** Existing explainability methods in deep learning primarily determine which spatial regions of the input image or a combination of regions led to a specific decision or contribute significantly to the network prediction (see Figure 1). There are two main approaches to identify regions of interest when using DNNs: Model-based methods and model agnostic methods. Model-based methods work based on the details of the specific architecture of a DNN. These methods examine the activations or weights of the DNN to find regions of importance. Grad-CAM and Layerwise Relevance propagation Samek et al. (2016) are examples of such methods. Attention-based methods Dosovitskiy et al. (2020) similarly identify important image regions. Model agnostic methods methods separate the explanations from the model which offers wide applicability. These methods (e.g., LIME Ribeiro et al. (2016)) manipulate inputs (e.g., pixels, regions or superpixels) and measure how changes in input images affect the output. If an input perturbation has no effect, it is not relevant to decision-making. In contrast, if a change has a major impact (e.g., changing the classification from melanoma to normal), then the region is important to the classification. SHapley Additive exPlanations (SHAP) Lundberg & Lee (2017) can assign each feature or region an importance value for a particular prediction. Note, however, the regions found by these algorithms do not necessarily correspond to intermediate concepts or diagnostic features that are known to experts or novices. Hence, while these algorithms are helpful to explain classifications of DNNs, they do not help training models that mimic humans when making predictions. In the evolving landscape of explainable AI, particularly in the medical domain, counterfactual explanations have gained prominence. A study by Metta et al. (2023) focuses on enhancing trust in medical diagnoses of skin lesions through transparent deep learning models. Dhurandhar et al. (2018) introduce a novel angle to counterfactual explanations by emphasizing pertinent negatives. Bodria et al. (2023) provide an extensive overview and benchmarking of various explanation methods for black-box models. Collectively, these studies highlight the necessity and methods of explainable models in medical diagnostics for greater acceptance and reliability.

Identifying regions of interest is also related to semantic segmentation Noh et al. (2015); Stan & Rostami (2022) which divides an image into segments that are semantically meaningful (e.g., separating moles from background skin in diagnosing melanoma or segmenting a clinical indicator from the rest of a mole). U-Nets Ronneberger et al. (2015) specifically have been found to be quite effective in solving segmentation tasks within medical domains. However, they do not indicate the importance of regions to overall classification, a key step in explaining model decision. The

major deficiency is that most segmentation image methods mostly segment based on spatial similarities and do not offer any explanation how these segments that are generated can be used for classification of the input image. In our works, we build an explainable architecture that solves segmentation tasks that localize clinically interpretable indicators on input images and then train a classifier sub-network to make decisions based on the identified indicators.

**Deep Learning for Melanoma Diagnosis**  Dermatology is one of the most common use cases of DL in medicine, with many existing works in the literature Codella et al. (2017); Sultana & Puhan (2018); Li & Shen (2018); Adegun & Viriri (2019); Kassani & Kassani; Naeem et al. (2020); Jojoa Acosta et al. (2021). Despite significant progress in DL, these methods simply train a DL on a labeled dataset, often binary labels for cancerous vs benign, using supervised learning. Despite being naive in terms the learning algorithms they use, these works lead to decent performances, comparable with expert clinicians. However, there is still room for improving explainability characteristics of these methods to convince clinicians adopting DL for melanoma diagnosis in practice. Only a few existing works have explored explainability of AI models for melanoma diagnosis. Murabayashi et al. Murabayashi & Iyatomi (2019b) use clinical indicators and benefit from virtual adversarial training Miyato et al. (2018) and transfer learning Isele et al. (2016) to train a model that predicts the clinical indicators in addition to the binary label to improve explainability. Nigar et al. Nigar et al. (2022) simply use LIME to study interpretability of their algorithm. Stieler et al. Stieler et al. (2021) use the ABCD-rule, an empirically-driven diagnostic approach of dermatologists, while training a model to improve interpretability. Shorfuzzaman Shorfuzzaman (2022) used meta-learning to train an ensemble of DNNs, each predicting a clinical indicator, to use indicators to explain decisions. These existing works, however, do not spatially locate the indicators. We develop and architecture that solves segmentation tasks to generate spatial masks on the input image to locate clinical indicators spatially through solving a weaky-supervised tasks. As a result, our architecture identifies clinical indicators and their spatial location on input images to explain particular decisions that it makes.

**Self-Supervised Learning for Data Augmentation**  We rely on intermediate-level annotations to solve segmentation tasks. The challenge is that annotating medical data is often expensive and existing annotated datasets are small. To over come this challenge, we rely on self-supervised learning (SSL) for data augmentation. SSL harnesses inherent patterns and relationships using unannotated data to enhance the robustness and generalization of models when acquiring large annotated datasets proves challenging. SSL has found notable traction in solving medical image analysis tasks. To name a few works, Chen et al. Chen et al. (2019) demonstrated the benefit of self-supervised learning for utilizing unannotated datasets in medical image analysis. Azizi et al. Azizi et al. (2021) demonstrate that using SSL can significantly increase classification rate of medical images. Tang et al. Tang et al. (2022) utilized SSL for extracting robust features that enhance medical image segmentation. We have also demonstrated that SSL can enhance model generalization in unsupervised domain adaptation scenarios Jian & Rostami (2023). These works underlie the potential of SSL in addressing data scarcity and enhancing the quality of medical analyses. In our work, we rely on SSL to improve the quality of features that are learned to preform the corresponding segmentation tasks.

## 3  Problem Formulation

Our goal is to develop a DL framework for melanoma diagnosis such that the model provides human-interpretable explanations behind its decisions. Most existing works for melanoma diagnosis based on DL consider that we have access to a single dataset that includes skin lesion images along with corresponding binary labels for cancerous vs benign cases Premaladha & Ravichandran (2016); Zhang (2017); Codella et al. (2017); Sultana & Puhan (2018); Li & Shen (2018); Adegun & Viriri (2019); Kassani & Kassani; Naeem et al. (2020); Jojoa Acosta et al. (2021). The standard end-to-end supervised learning is then used to train a suitable DNN, e.g., a convolutional neural network (CNN). However, as explained, this simple baseline does not lead to training a model with human interpretable explanations. To overcome this challenge, we assume that we have access to a dataset annotated with intermediate information that are in the forms of spatial masks that visualize clinical indicators on the input images (see Figure 3, where the second column visualizes such masks). Our goal would be developing a DNN architecture that not only learns to classify skin lesion images, but also localizes clinical indicators and then uses them for decision making, similar to clinicians.

To train explainable DL architectures, we can rely on public datasets where images are annotated with clinically plausible indicators Codella et al. (2019); Tschandl et al. (2018). These indicator commonly are used by dermatologists. Dermatology residents are often trained to diagnose melanoma based on identifying them. We try to mimic this two-stage diagnosis protocol that clinicians use by training the model to first predict and localize the indica-

tors as intermediate-level abstractions and then use them for predicting the downstream binary diagnosis label. Let $D^L = \{\boldsymbol{x}_i, y_i, (\boldsymbol{z}_{ij})_{j=1}^d\}_{i=1}^M$ denotes this dataset, where $\boldsymbol{x}_i$ and $y_i$ denote the images and their binary diagnostic labels. Additionally, $\boldsymbol{Z}_{ij}$ denotes a feature mask array with the same size as the input image, where for each $1 \leq j \leq d$, the mask denotes the spatial location of a clinically interpretable indicator, e.g., pigmented network, on the input image in the form of a a binary segmentation mask (see Figure 3 for instances of such a dataset). In our formulation, we refer to this dataset as *Dataset A* which can be used to learn predicting and locating clinical indicators.

A naive idea is for training an explainable model is to use a suitable architecture and train one segmentation model per each indicator, e.g., U-Net Ronneberger et al. (2015), to predict indicator masks given the input. Previously, this idea has been used for training explainable ML models in medical domains Sharma et al. (2022). In our problem, we can use one U-Net for each of the $d$ indicators and train them using *Dataset A*. Hence, we will have $d$ image segmentation models that determine spatial locations of each indicator for input images. However, there are two shortcomings that makes this solution non-ideal. First, we will still need a secondary classification model to determine the diagnosis label from the indicators Murabayashi & Iyatomi (2019b) and coupling it with the segmentation models is not trivial. More importantly, the size of the *Dataset A* may not be large enough for training segmentation models. The challenge in our formulation is that privacy concerns and high annotation costs of medical image datasets limits the size of publicly available instances of *Dataset A* that are well-annotated with indicator masks. Moreover, only a subset of instances contains a particular type of indicator which makes the dataset sparse. Since we likely will encounter the challenge of attribute sparsity, we likely will face overfitting during testing stage. Increasing the size of *Dataset A* can be helpful but note that preparing *Dataset A* is a challenging task because existing medical records rarely include instances of indicator masks. Hence, a dermatologist should determine the absence and presence of each indicator and locate them on images in addition to a binary diagnosis label which is a time-consuming task. Our idea is to benefit from an additional dataset that is easier to prepare to make learning from a small instance of *Dataset A* feasible.

Fortunately, there are larger publicly available datasets that are coarsely annotated, where only the binary diagnostic label is accessible. We aim to leverage such a large coarsely-annotated dataset to solve the corresponding segmentation tasks. Let $D^{UL} = \{\boldsymbol{x}_i'\}_{i=1}^N$ denote such a dataset, where $\boldsymbol{x}_i'$ denotes an input image and $M << N$. We refer to this dataset as *Dataset B* in our formulation. Preparing *Dataset B* is much easier than preparing *Dataset A* because we only need to go though existing medical records to prepare *Dataset B* according to the filed diagnosis. We benefit from both *Dataset A* and *Dataset B* to train an architecture that learns to identify and localize clinically relevant indicators and then predict the diagnostic labels. In other words, we use *Dataset B* as a source of knowledge to overcome the challenge of having a small *Dataset A*. Note that we cannot benefit from unsupervised domain adaptation because we want to transfer knowledge from the unannotated domain to the annotated domain. For this reason, we use SSL for extracting robust features. Although *Dataset B* is not attributed finely, it is similar to *Dataset A* and transferring knowledge between these two datasets is feasible. We formulate a weakly supervised learning problem for this purpose. Specifically, we use SSL to train an encoder that can better represent input images using *Dataset B*. SSL would enable the encoder to extract features that can be used to locate the indicators. Additionally, we train multiple encoders to separately learn each indicator, so that we can apply unique operations to each encoder and improve performance. Finally, we concatenate the output feature vectors of these encoders and use the resulting feature vector to predict the binary labels $y_i$, making the model's prediction process similar to an expert.

## 4 Proposed Architecture and Algorithm

We outline our proposed architecture and its key components, emphasizing the roles of each module. The primary goal is to train the Bio-U-Net architecture to ensure the Grad-CAM heatmaps from ResNet's last convolution layer align closely with the ground truth mask.

### 4.1 Overview of the network architecture

Figure 2 visualizes the proposed architecture. In short, we first train a base network for diagnostic classification and then generate a heatmap using the final convolution layer. We then use the generated heatmap as inputs to a segmentation module to localize each melanoma indicator. Our architecture, named Bio-U-Net, consists of four subnetworks:

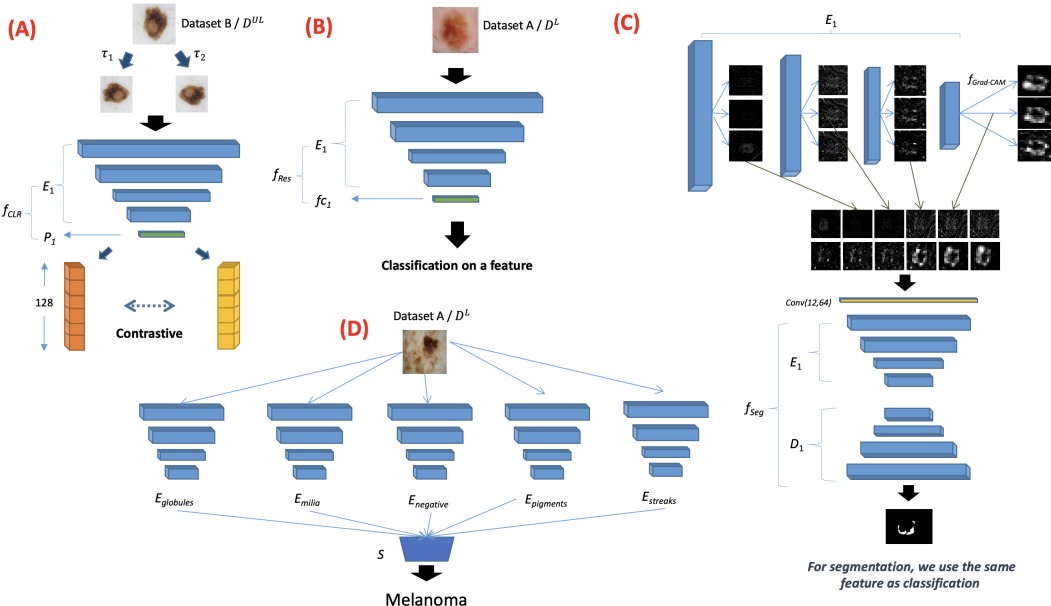

Figure 2: Proposed architecture for explainable diagnosis of melanoma: the architecture is trained to simultaneously classify skin lesion pictures using a CNN classifier and learn to localize melanoma clinical indicators on the input image using a U-Net-based segmentation network that receives its input from the heatmaps that encode attention of the classification subnetwork.

(A) Our foundational network, referred to as subnetwork(1), is a ResNet50 backbone, pretrained on ImageNet. Since we benefit from self-supervised learning to train subnetwork(1), we append a projection head at the final ResNet50 layer to adapt the base architecture. The entire architecture of subnetwork(B) is denoted as $f_{\text{CLR}}(\cdot)$. Formally, the network's operation can be expressed as $f_{\text{CLR}}(\boldsymbol{x}_i') = P_1(E_1(\boldsymbol{x}_i'))$, where $\boldsymbol{x}_i' \in D^{UL}$. In this expression, $E_1(\cdot)$ signifies the feature extraction capacity of ResNet50, and $P_1(\cdot)$ represents our projection head, which consists of two fully connected layers that outputs a feature vector.

(B) For diagnostic classification, we employ a separate ResNet50 network, referred to as subnetwork(2). This subnetwork utilizes the trained feature extractor, $E_1(\cdot)$, from subnetwork(1). Subnetwork(2) can be depicted as $f_{\text{Res}}(\boldsymbol{x}_i) = fc_1(E_1(\boldsymbol{x}_i))$, where $\boldsymbol{x}_i \in D^L$ and $fc_1(\cdot)$ is a fully connected layer for label prediction.

(C) We introduce a segmentation subnetwork, termed *subnetwork(3)*, grounded on the U-Net architecture for pinpointing clinical indicators. Instead of the standard U-Net encoder, we incorporate the trained encoder $E_1(\cdot)$ from *subnetwork(2)*. *Subnetwork(3)* is described by $f_{\text{Seg}}(H)$, where $H$ consists of 12 heatmaps. Each heatmap is generated from a bottleneck unit of ResNet50 using Grad-CAM, considering that all three convolutional layers within such a unit produce analogous localization maps with Grad-CAM. Hence, we extract one representative heatmap from each bottleneck unit, leading to 3 heatmaps per block and a total of 12 for the entire ResNet50 architecture. The decoder $D_1(\cdot)$ mirrors $E_1(\cdot)$ and crafts indicator masks, aiding precise localization, with encoder-decoder connections similar to the orignal U-Net architecture.

(D) Subnetwork(4): After training a subnetwork(3) for all melanoma indicators, all the encoders — $E_{\text{pigment}}, E_{\text{globule}}, E_{\text{negative}}, E_{\text{streaks}}$, and $E_{\text{milia}}$ — are frozen to ensure their weights remain unchanged (for a discussion on these indicators, please check the Experimental Setup section). For every image $\boldsymbol{x} \in D^L$, the encoded vectors from each of the five encoders are computed and concatenated as:

$$\boldsymbol{v}(\boldsymbol{x}) = [E_{\text{pigment}}(\boldsymbol{x}), E_{\text{globule}}(\boldsymbol{x}), E_{\text{negative}}(\boldsymbol{x}), E_{\text{streaks}}(\boldsymbol{x}), E_{\text{milia}}(\boldsymbol{x})].$$

This concatenated vector $\boldsymbol{v}(\boldsymbol{x})$ is then fed to a softmax layer $S(\cdot)$ which is exclusively trained:

$$S(\boldsymbol{v}(\boldsymbol{x})) = \text{final classification of } \boldsymbol{x}.$$

In short, our architecture learns to classify the input images. Then, we use Grad-CAM to generate 12 attention heatmaps. These heatmaps do not necessarily show interpretable indicators, but are useful for classification which means that they should have correlations with features that expert clinicians use. Our idea is to feed these heatmaps to the segmentation subnetworks and generate the indicator biomarker localization masks using GRAD-CAM heatmaps. As a result, the architecture learns to generate explanations along with diagnosis labels. We can see that our full architecture can be trained using only *Dataset A* but as we discussed using SSL is necessary to mitigate overfitting.

## 4.2 Self-Supervised Learning for Bio-U-Net

We aim to train the Bio-U-Net architecture such that the Grad-CAM maps produced from the last convolution layer of $E_1$ closely align with the ground truth mask. Note that the number of annotated images for certain biomarker indicators (e.g., the pigment network) is limited and other indicator appear more frequently a in existing datasets. This prominence can introduce biases, potentially overshadowing the detection of less frequent indicators. To tackle these challenges and to utilize both *Dataset A* and *Dataset B* effectively, we use SimCLR Chen et al. (2020) to benfit from self-supervised learning to enhance visual representations in two primary aspects:

1. **Diverse Feature Representations:** SimCLR helps our model to recognize diverse image patterns. By contrasting various transformations of the same image, the model captures the subtleties of less common indicators while not solely focusing on predominant diagnostic indicators.

2. **Leveraging Unannotated Data:** SimCLR capitalizes on the valuable information in the unannotated dataset. As a result, the model learns richer representations, bolstering its capability to identify the indicators.

As visualized in Figure 2, two independent data augmenters, $T_1(\cdot)$ and $T_2(\cdot)$, chosen randomly from transformations that include rotation, scaling, cropping, and flipping, generate augmented versions of *Dataset B* samples. This augmentation is crucial for determining the contrastive learning loss. Each image $\boldsymbol{x}'_i \in D^{UL}$ yields two unique augmented images after processing through the augmenters. These images are fed into the shared encoder $E_1(\cdot)$ and projection head $P_1(\cdot)$ to produce two 128-length features. Each training cycle processes a minibatch of $N$ input images, creating $2N$ augmented images in total. We treat each pair of augmented images as positive samples, whereas the other $2(N-1)$ are considered negative samples. The contrastive loss is adopted as our semi-supervised loss:.

$$L_{CLR} = -\log \frac{\exp(sim(f_{CLR}(T_1(x_i)), f_{CLR}(T_2(x_j)))/\tau)}{\sum_{k=1}^{2N} \exp(sim(f_{CLR}(T_1(x_i)), f_{CLR}(T_2(x_k)))/\tau)}, \tag{1}$$

where, $sim(u,v) = \frac{u^T v}{\|u\|\|v\|}$, $k \neq i$, and $\tau$ is a temperature parameter. After training the encoders on *Dataset B*, the obtained knowledge can be transferred to *Dataset A*. Our complete training pipeline is presented in Algorithm 1.

## 4.3 Bio-U-Net Baseline Training and Feature-driven Attention Heatmaps

The baseline architecture of Bio-U-Net consists of two primary subnetworks: $f_{Res}(\cdot)$ and $f_{Seg}(\cdot)$.

**Classification and Localization:** we first train $f_{Res}(\cdot)$ for the task of skin lesion classification using a supervised learning strategy. This network is adept at predicting diagnostic labels with a high accuracy. To identify regions of the input image that correspond to the model's decision, we employ Grad-CAM. While Grad-CAM can compute the gradient of the classification score with respect to the last convolutional feature map, it often highlights a broader area that doesn't always align with the detailed annotations of experts which we try to improve upon in our work.

**Fusion of Semantic and Structural Information:** wet meld the low-level details of an image, such as boundaries or textures of objects, with high-level semantic information, like the overall context of the object. This fusion ensures that the network not only focuses on the overall lesion area but emphasizes regions that align more closely with expert annotations. Since early convolutional layers can provide attention to low-level details and high-level layers give more semantic context, we benefit from all layer Grad-CAM heatmaps within the network. These maps illustrate how the network attends to the image at various abstraction levels. Merely averaging these attention maps might not provide precise information regarding the location of a specific indicator. Hence, we used the binary masks provided by experts and reconstructed these masks using the $f_{Seg}(\cdot)$ subnetwork. Note that we optimized a model for each diagnostic indicator separately, ensuring detailed attention to each specific one, resulting in accurate localization and

subsequent interpretation. In devising this strategy, we adopted a new encoder $E_3(\cdot)$ which mimics the architecture of $E_1(\cdot)$ to avoid interference with the parameters of $E_1(\cdot)$. For every training image, we use a bottleneck block within Grad-CAM to generate attention maps, giving us a set of localization masks, offering attention information at different abstraction levels. This process can be represented as $f_{Grad\_CAM}(b_i, j)$, where $b_i$ is the bottleneck block and j is the label index. As a result of our training pipeline, we use the attention maps created by the network to classify images, to reconstruct semantic maps for the clinical indicators. Hence, when our model classifies an image to be cancerous, it also provides masks that locate the relevant clinical indicators on the input image that has led to the model's decision.

### 4.4 Refining Heatmap for Diagnostic Clarity

Refining the heatmaps generated by the model is of important for having accurate predicted masks. Improved heatmaps not only elucidate model decisions but also augment its diagnostic capability. The quality of these heatmaps is pivotal for accurately pinpointing the regions of interest that underpin the model's predictions. We introduce a novel strategy by feeding 12 distinct heatmaps into the model. Originating from different layers of the network, these heatmaps encapsulate a plethora of features, especially capturing those minute details that are often glossed over by classic explainability techniques. By emphasizing on these granular aspects, our model demonstrates an enhanced sensitivity to features that can be crucial for precise diagnostic evaluations. Since inputting a multitude of heatmaps might scatter the model's focus, we employ the soft dice loss:

$$L_{SoftDice} = 1 - \frac{2 \sum_{Pixels} y_{true} y_{predict}}{\sum_{Pixels} y_{true}^2 + \sum_{Pixels} y_{pred}^2} \quad (2)$$

---

**Algorithm 1** Proposed Architecture Training Approach

---

**Inputs** $(x_i, y_i)_{i=1}^N \in D^L$, $(x'_i, y'_i)_{i=1}^M \in D^{UL}$, $(b_i)_{i=1}^{12}$
1: Initialize subnetworks:
2:      $f_{CLR}(x'_i) = E_1(P_1(x'_i))$
3:      $f_{Res}(x_i) = fc_1(E_1(x_i))$
4:      $f_{Seg}(H) = D_1(E_1(H))$
5: Initialize $f_{Grad\_CAM}(b_i, j)$ where $b_i \in E_3$
6: Define $T_1, T_2$ : two separate data augmentation operators
7: **while** j < d **do**
8:      **while** Not stop **do**
9:          Sample batch $B_1 = x'_i \in D^{UL}$
10:          Generate $f_{CLR}(T_1(B_1))$ and $f_{CLR}(T_2(B_1))$
11:          Update $E_1(\cdot)$ and $P_1(\cdot)$ using $L_{CLR}$
12:          Sample batch $B_2 = \{(x_i, y_i) \in D^L\}$
13:          Compute $f_{Res}(B_2)$ and update $E_1(\cdot)$, $fc_1(\cdot)$ using $L_{BCE}$
14:          Load optimal $\theta$ for $E_3$, use $f_{GradCAM}(b_i, j)$ to compute $h_i$
15:          Assemble $H = (h_i)_{i=1}^{12}$
16:          Generate $f_{Seg}(H)$ and update $E_1(\cdot)$, $D_1(\cdot)$ by optimizing the $L_{SoftDice}$ loss
17:      **end while**
18: **end while**
19: Load five best encoder parameters onto $E_{pigment}, E_{globule}, E_{negative}, E_{streaks}, E_{milia}$
20: Concatenate the outputs from the 5 encoders
21: Train a logistic regression model using the concatenated outputs

---

Distinct from conventional binary losses, the soft dice loss employs the predicted probabilities and facilitates a smoother gradient flow, nudging the model to be more receptive to subtle intricacies. Our objective isn't to perfect segmentation outcomes using these heatmaps, but to hone the model's attention to essential features. By synergizing the multi-heatmap input strategy with the soft dice loss, we generate sharper and enhanced Grad-CAM heatmaps. This refined visualization proffers diagnostic lucidity, offering a transparent window into the model's decision-making.

## 5 Experimental Validation

We validate our architecture using real-world datasets. Our implementation code is provided as a supplement.

### 5.1 Experimental Setup

**Datasets** We used the ISIC dataset Codella et al. (2019); Tschandl et al. (2018). The dataset is a real-world repository of dermatoscopic images, making it ideal for our experiments.

*Dataset A* We extracted data from Task 2 of the ISIC 2018 dataset. This task is designed to detect five critical dermoscopic attributes: pigment network, negative network, streaks, milia-like cysts, and globules. The importance of these attributes is underscored by their wide application in clinical melanoma detection processes. For a comprehensive understanding of these biomarkers, readers can refer to the detailed ISIC 2018 documentation Codella et al. (2019); Tschandl et al. (2018). Dataset A comprises 2594 images, each having binary labels for melanoma diagnosis. The distribution of these attributes is represented in Table 1 which reveals the sparse nature of the attributes.

*Dataset B* We use Task 3 of ISIC 2019 which is considerably larger with 25331 images. Each image in *Dataset B* has only binary diagnosis labels, without detailed annotations pertaining to the specific melanoma indicators.

| Dermoscopic Attribute | Images with Attribute (Non-empty Masks) | Images without Attribute (Empty Masks) |
|---|---|---|
| globules | 603 | 1991 |
| milia_like_cyst | 682 | 1912 |
| negative network | 190 | 2404 |
| pigment network | 1523 | 881 |
| streaks | 100 | 2494 |

Table 1: Distribution analysis for ISIC 2018 Task 2 dataset, detailing images that possess or lack specific dermoscopic attributes.

**Baselines for Comparison:** While the ResNet50 subnetwork forms the backbone of Bio-U-Net, the modifications and layers introduced in Bio-U-Net are tailored to optimize both accuracy and interpretability in tandem. We juxtapose Bio-U-Net with ResNet50 when common interpretability techniques is applied on a ResNet50 trained for classification. We relied on the class activation mapping (CAM) methods, including, Layer-CAM, Grad-CAM, and Grad-CAM++. The utility of LIME on melanoma datasets necessitated its inclusion in our comparisons. Despite a rich literature on adopting AI for diagnosing melanoma, there are not many existing methods that offer interpretability. We could only compare against VAT Murabayashi & Iyatomi (2019a) which is a more advanced method and is the most similar method to our work. For all these techniques, interpretability was performed on the last layer of ResNet50.

**Evaluation Metrics:** we aim for accurate pinpointing of melanoma indicators within images. We use the Continuous Dice metric Shamir et al. (2019) because it's effective at comparing how closely our generated indicator masks match the human annotated ones. Additionally, we measure classification accuracy to understand how well we're diagnosing melanoma. Our evaluation balances both the clarity of our model's decisions and its ability to diagnose accurately.

**Implementation Details** For details about the algorithm implementation, optimizations hyperparameters, and the used hardware, please refer to the supplementary material. Our code also includes all details for implementation.

## 5.2 Performance Analysis

We first analyze the influence of melanoma indicators on model diagnosis efficacy. Upon training, the ResNet50 subnetwork and the Bio-U-Net architecture achieve classification accuracies of $76\%$ and $82\%$, respectively. We observe that Bio-U-Net's embedded subnetworks for segmentation and localization not only has the potential to enhance explainability, but also demonstrably elevate classification performance. This observation is not surprising because the clinical protocols can be considered a gold standard diagnosis procedure that have emerged during decades of clinical procedures. We conclude that incorporating human expert knowledge into the deep learning pipeline can improve performance of deep learning in complicated tasks for which the size of the annotated training dataset is not large.

Table 2 provides a comparison for the efficacy of localizing the five melanoma indicators, benchmarked using the Continuous DICE metric. Figure 3 provides localization results for a sample of input images to offer the possibility of visual inspection for the possibility of a more intuitive comparison. Additional examples are also provided in the Appendix. A close inspection of the results underscores several key observations:

| | globules | milia_like_cyst | negative | pigment | streaks |
|---|---|---|---|---|---|
| $f_{Res}(\cdot)$ + Grad-Cam | 14.21 | 0.0 | 15.78 | 41.03 | 5.16 |
| $f_{Res}(\cdot)$ + Grad-Cam++ | 7.95 | 0.67 | 13.67 | 26.85 | 3.5 |
| $f_{Res}(\cdot)$ + Layer-Cam | 14.53 | 0.0 | 15.89 | 40.67 | 5.83 |
| $f_{Res}(\cdot)$ + Lime | 3.36 | 1.65 | 4.00 | 13.26 | 2.00 |
| VAT Murabayashi & Iyatomi (2019a) | 2.05 | 0.76 | 2.33 | 19.36 | 1.5 |
| Bio-U-Net | 15.16 | 2.33 | 20.89 | 32.79 | 14.17 |

Table 2: Melanoma indicator localization performance comparison based on the Continuous DICE metric. Higher values indicate better performance.

- **CAM-Based Explainability Methods:** The CAM-based methods, applied atop ResNet50, have a consistent performance pattern. For instance, 'globules' indicator is identified at 14.21%, 7.95%, and 14.53% rates using

Grad-CAM, Grad-CAM++, and Layer-CAM, respectively. Their general tendency to focus on central regions yields almost circular areas of attention. The circularity suggests an overemphasis on dominant features, potentially sidelining subtle yet crucial indicators, leading to deviation from human-centered interpretability.

- **Lime:** yields a Continuous DICE score of 3.36% for 'globules' and 1.65% for 'milia_like_cyst'. Its heatmap is scattered with white pixels, indicating a widespread attribution which, although granular, is less focused. This diffused pattern poses challenges when pinpointing specific regions pivotal for melanoma diagnosis.

- **VAT:** presents an innovative approach towards predicting explainable melanoma indicators using the 7-point checklist, especially when faced with a limited number of labeled data. This method offers potential improvements in melanoma diagnosis accuracy, even rivaling expert dermatologists. When using Grad-CAM to observe its attention detection regions, VAT's performance seems suboptimal for certain features, manifesting scores such as a mere 2.05% for 'globules' and 19.36% for 'pigment'. This observation suggests that while VAT's approach is unique and holds promise, there are areas in which its feature detection capability, especially in the heatmaps generated from the final layer of ResNet50, could be enhanced. This is particularly evident for more dispersed or nuanced features like 'streaks', where it only managed a score of 1.5%.

- **Bio-U-Net:** Bio-U-Net's prowess stands out. It registers scores of 15.16% for 'globules', a marked 2.33% for 'milia_like_cyst'—a clear rise from the near-zero performance by some CAM methods—and an impressive 20.89% for 'negative'. The architecture reliably zeroes in on dispersed attention regions, resulting in cohesive and pinpointed heatmaps. These figures, especially the leap in scores for indicators like 'negative' and 'streaks' to 20.89% and 14.17% respectively, underline Bio-U-Net's adeptness at detecting indicators.

These analytical findings accentuate the pivotal role of integrating human-interpretable annotations into deep learning training pipelines. They advocate for DNN architectural designs that not only enhance explainability but also leverage these annotations to optimize the model's overall efficacy for the designated tasks.

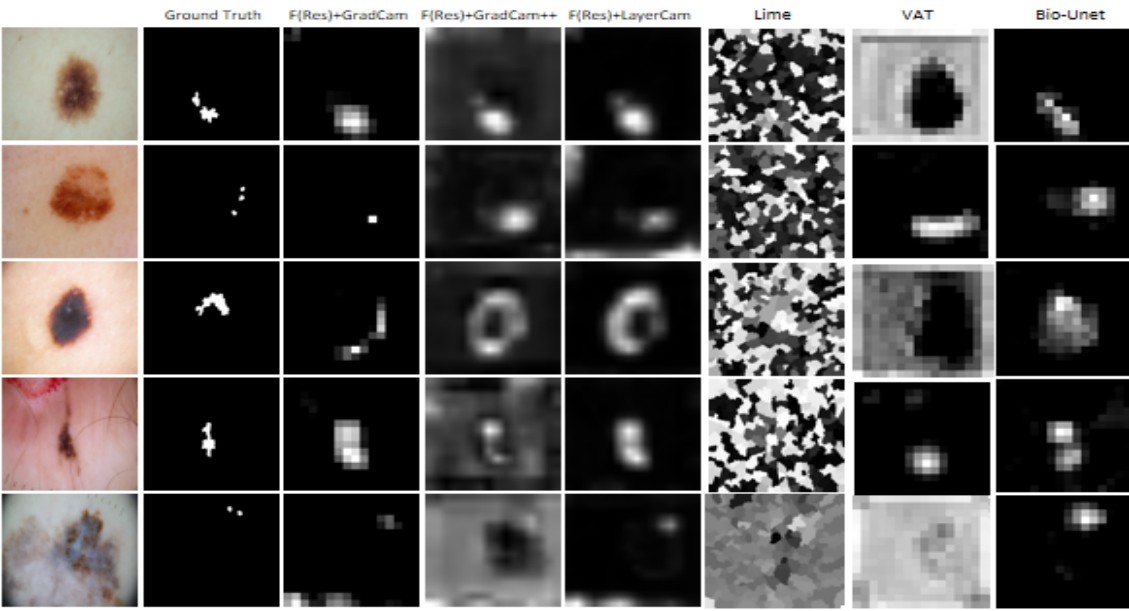

Figure 3: Localization of indicators for samples of dermatoscopic images. Presented from top to bottom is a sample input image accompanied by localization maps crafted by clinicians for several biomarkers indicators: globules, milia like cyst, negative network, pigment network, and streaks. From left to right, visualizations cover the input image, the ground truth mask of the indicator, and masks produced by Grad-CAM, Grad-CAM++, LayerCAM, Lime, VAT, and Bio-U-Net. CAM-based feature maps, underpinned by a Resnet50 backbone trained for classification, reveal that Bio-U-Net generated masks are aligned with human-created masks.

### 5.3 Ablative and Analytic Experiments

We provide experiments to offer a deeper insight about our proposed architecture. We first study the effect of the architecture components on the downstream performance. We then offer experiments with alternative possibilities to demonstrate that our design is optimal.

**Ablative experiments on the components of the proposed architecture:** We conducted an ablation study to investigate the contribution of each component of Bio-U-Net on the localization performance. Table 3 presents the results of our ablative study when the subnetworks are removed from the architecture. We observe that when the subnetwork $f_{CLR}(\cdot)$ is removed, the localization results for the "streaks" and "negative network" indicators were reduced. This observation was expected because "streaks" and "negative network" indicators pertain to only a small number of samples and appear in a scattered and discontinuous manner in the input images. We conclude that SSL is extremely helpful for localizing infrequent indicators that appear in a scattered and discontinuous manner in the input images.

| | $f_{Seg}$ | $f_{CLR}$ | globules | milia_like_cyst | negative | pigment | streaks | Mel |
|---|---|---|---|---|---|---|---|---|
| $f_{Res}(\cdot)$ | | | 14.21 | 0.0 | 15.78 | 41.03 | 5.16 | 0.76 |
| $f_{Res}(\cdot) + f_{Seg}(\cdot)$ | ✓ | | 15.21 | 0.5 | 17.78 | 39.58 | 3.83 | 0.76 |
| $f_{Res}(\cdot) + f_{CLR}(\cdot)$ | | ✓ | 15.11 | 1.0 | 17.44 | 39.53 | 12.67 | 0.80 |
| Bio-U-Net | ✓ | ✓ | 15.16 | 2.33 | 20.89 | 32.79 | 14.17 | 0.82 |

Table 3: Ablative study on the importance of subnetworks of Bio-U-Net on identifying indicators.

Figure 4 also presents generated masks from the ablative study on the subnetworks using a number of samples. Rows represent different indicators, while columns denote various network configurations in line with Table 3. We observe that for the case of "globules" (first row), Bio-U-Net produces a sharp boundary compared to when either $f_{CLR}(\cdot)$ or $f_{Seg}(\cdot)$ is omitted. Both of these singular removals manifest in suboptimal mask outcomes. In the case of "milia_like_cyst" (second row), omission of $f_{CLR}(\cdot)$ leads to a mask that is blurred and offset. Without $f_{Seg}(\cdot)$, the mask aligns closely with the ground truth. Bio-U-Net's rendition aligns almost perfectly with the ground truth. For "negative network" (third row), masks post $f_{CLR}(\cdot)$ removal almost vanish, highlighting the component's importance. Omitting just $f_{Seg}(\cdot)$ yields an accurate mask. In the "pigment network" instance (fourth row), characterized by scattered indicators, we observe that without $f_{Seg}(\cdot)$, the network's attention is on five specific dots. After $f_{CLR}(\cdot)$ removal, focus narrows to two primary areas, indicating $f_{CLR}(\cdot)$'s role in broadening the network's attention to dispersed regions. Finally, for "streaks" (fifth row), we see that omitting $f_{CLR}(\cdot)$ results in a smaller, concentrated mask. Without $f_{Seg}(\cdot)$, there's a larger, central-focused mask. Bio-U-Net divides its attention uniformly, mirroring the ground truth closely. In summary, contrasting different configurations accentuates each component's critical role in Bio-U-Net to generate masks that are more similar to masks generated by expert dermatologists.

**Impact of updating $f_{Seg}(\cdot)$ within one training epoch:** A natural question for our optimization procedure is how often $f_{Seg}(\cdot)$ should be updated during each training epoch for improved performance. Figure 5 shows how segmentation performance metrics change when we update $f_{Seg}(\cdot)$ more times across attributes. The first column in the figure gives the input data, the second column shows results from Bio-U-Net, and the next columns show results when we repeat $f_{Seg}(\cdot)$ two, three, four and five times, respectively. Looking at the 'streaks' attribute, Bio-U-Net (in the second column) pinpoint two main areas quite well. We conclude that additional updating of $f_{Seg}(\cdot)$ is not necessarily helpful.

For a thorough analysis, Table 4 provides the impact of refining $f_{Seg}(\cdot)$ on indicator localization performance across all dataset samples. We observe that updating $f_{Seg}(\cdot)$ once can improve results for four of the five clinical indicator. But the downside is that the accuracy value for diagnosing melanoma drops. For instance, the best score for "pigment network" happens after the fourth repeat, but repeating also lowers the score for "streaks". This observation indicates that additional updates can be helpful for some features but not for others. For this reason, we update $f_{Seg}(\cdot)$ just once per iteration to keep our procedure consistent, simple, and straightforward.

**Using five separate encoders vs. standard multi-task learning** We study whether using multi-task learning (MTL) can improve the results. Table 5 provides results for MTL. In this table, *Bio-U-Net* uses an encoder that predicts one intermediate feature and the Melanoma label. *Bio-U-Net-(Two-tasks)* and *Bio-U-Net-(Five-tasks)* let the encoder predict two and five intermediate indicators, respectively. *Bio-U-Net-(Six-tasks)* lets the encoder predict all five intermediate

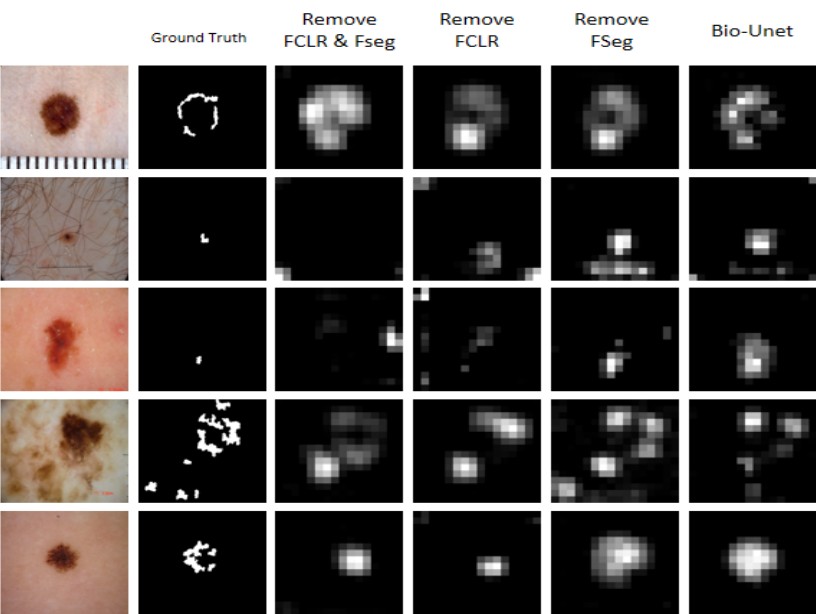

Figure 4: Generated masks for the ablative study on subnetworks: from top to bottom are samples of globules, milia_like_cyst, negative network, pigment network, streaks. From left to right are input image, ground truth mask, and mask generated by $f_{Res}(\cdot)$, $f_{Res}(\cdot) + f_{CLR}(\cdot)$, $f_{Res}(\cdot) + f_{Seg}(\cdot)$, and Bio-U-Net, following Table 3.

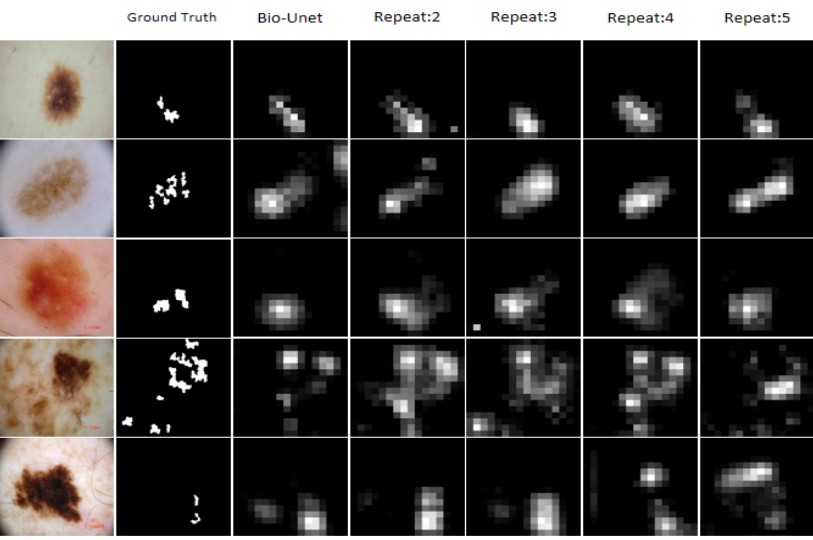

Figure 5: Impact of updating $f_{Seg}(\cdot)$: the columns, from left to right, display ground truth, one-repeated (Bio-U-Net), repeated once, repeated twice, repeated three times, and repeated four times. Vertically, the rows present samples of globules, milia_like_cyst, negative network, pigment network, and streaks.

features and also adds the Melanoma label. From the table, it's clear that adding the Melanoma label helps increasing AUC to 84% and the scores for "pigment" and "streaks". We conclude that the melanoma label is important to improve performance. However, when we predict five or six tasks at once, the Continuous Dice Coefficient (cDC) for some indicators becomes less. This observation indicates that when a model is trained to solve a number of tasks in an MTL setting, it might not perform better on all tasks. One reason might be that when we optimize $f_{Seg}(\cdot)$ for each indicator in an MTL setting, the tasks might interfere with each other during training, leading to negative transfer across the tasks. Hence, we decided to use a separate encoder for each intermediate feature to ensure indicators do not interfere with each other. As a result, we observed improved cDC values, justifying our design choice.

|  | globules | milia_like_cyst | negative | pigment | streaks | Melanoma |
|---|---|---|---|---|---|---|
| $f_{Res}(\cdot)$ | 14.21 | 0.0 | 15.78 | 41.03 | 5.16 | 0.76 |
| Bio-U-Net (One Repeat) | 15.16 | 2.33 | 20.89 | 32.79 | 14.17 | 0.82 |
| Repeat 2 | 14.89 | 2.67 | 21.22 | 37.92 | 15.50 | 0.80 |
| Repeat 3 | 14.84 | 0.16 | 16.33 | 39.01 | 12.83 | 0.82 |
| Repeat 4 | 13.68 | 1.00 | 19.00 | 43.19 | 7.93 | 0.81 |
| Repeat 5 | 13.26 | 1.83 | 16.89 | 38.04 | 11.5 | 0.80 |

Table 4: Evaluation of localization precision for the five clinical indicators using the Continuous Dice Coefficient (in percentage).

|  | globules | milia_like_cyst | negative | pigment | streaks | Melanoma |
|---|---|---|---|---|---|---|
| $f_{Res}(\cdot)$ | 14.21 | 0.0 | 15.78 | 41.03 | 5.16 | 0.76 |
| Bio-U-Net | 15.16 | 2.33 | 20.89 | 32.79 | 14.17 | 0.82 |
| Bio-U-Net-(Two-tasks) | 14.16 | 1.16 | 20.22 | 37.88 | 14.33 | 0.84 |
| Bio-U-Net-(Five-tasks) | 15.89 | 0.5 | 19.44 | 15.64 | 10.67 | 0.82 |
| Bio-U-Net-(Six-tasks) | 14.68 | 1.5 | 19.33 | 23.21 | 11.83 | 0.83 |

Table 5: Comparison of accuracy for five clinical indicators using MTL. The table shows Continuous Dice Coefficient percentages.

**Effect of the mask generation threshold on the localization performance** A major parameter that affects the localization perofrmnace significantly is the process to generate binary masks from the architecture output. We study the effect of the threshold parameter value on the localization performance in Table **?**. We observe that in the case of "Globus" indicator, performance remains consistently acceptable across diverse thresholds which underlines Grad-CAM's efficacy for localizing this attribute. Such resilience indicates that the feature's spatial localization remains largely unaffected by threshold perturbations. However, the challenges of detecting the minuscule cysts emerge in the case of "Milia". Higher thresholds effectively suppress less relevant activations. This observation is substantiated by the table; average scores from the 0.5 to 0.9 thresholds outpace those between 0.0 and 0.4. This observation suggests that Milia's dispersed nature results in a diffused attention map. Hence, selecting the right threshold can therefore enhance model fidelity. In the case of "Negative" indicator, optimal performance was obtained at the threshold value of 0.6, emphasizing the benefits of focusing on dominant heatmap regions for this feature. This observation implies that the most influential regions correlate well with expected feature representations. In the case of `Pigment" indicator, notable peaks are observed at threshold values 0.2 and 0.5 which suggests that pruning suboptimal activations augments the clarity of pigment-centric localization. Finaly, for the "Streaks" indicator, given its potential fragmented nature, streaks present a formidable challenge for gradient-based localization tools like Grad-CAM. This empirical exploration suggests an optimal performance at the 0.6 threshold value. Note also that leveraging Grad-CAM visualizations from the final layer of ResNet assured consistent outcomes, minimizing significant deviations from the ground truth. A delicate balance between the intrinsic feature characteristics and heatmap preprocessing emerges as pivotal. In summary, this experiment provides the nuanced relationship between specific feature attributes and their heatmap representations. The decision to use a threshold of 0.6 in the actual experiment was guided by the varied behavior exhibited by different attributes, underscoring the complexity inherent in each feature.

| Thresholds | 0.0 | 0.1 | 0.2 | 0.3 | 0.4 | 0.5 | 0.6 | 0.7 | 0.8 | 0.9 |
|---|---|---|---|---|---|---|---|---|---|---|
| Globus | 15.16 | 15.21 | 14.78 | 15.31 | 14.05 | 13.78 | 15.26 | 15.63 | 14.73 | 17.00 |
| Milia | 2.33 | 2.29 | 4.29 | 1.01 | 0.06 | 3.85 | 3.84 | 3.79 | 2.67 | 4.19 |
| Negative | 20.89 | 21.22 | 16.00 | 23.44 | 14.11 | 18.22 | 22.22 | 20.66 | 15.88 | 18.44 |
| Pigment | 32.79 | 38.26 | 33.97 | 31.17 | 33.28 | 38.67 | 32.47 | 34.73 | 35.95 | 32.78 |
| Streaks | 14.17 | 14.33 | 12.16 | 13.83 | 13.00 | 14.33 | 18.66 | 14.50 | 15.16 | 13.33 |

Table 6: Dice coefficients across various thresholds for different features. The results reveal that each feature can have varying performance across different thresholds. The reported values are Continuous Dice Coefficient percentages.

## 6 Conclusions

We developed an architecture for explainable diagnosis of melanoma using skin lesion images. Our architecture is designed to localize melanoma clinical indicators spatially and use them to predict the downstream diagnosis label. As a result, it performs the task similar to a clinician, leading to a human interpretable decision-making process. We benefited from self-supervised learning to address the challenge of annotated data scarcity for our task because it requires coarse annotations with respect to clinical indicators. Experimental results demonstrate that our model is able to generate localization masks for identifying clinical biomarkers and generates more plausible and localized explanations compared to existing classification architectures. Future works include further verification by clinicians and extension to learning settings with distributed data and extensions to other similar clinical applications.

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

## A  Implementation Details

We provide details of our experimental implementation.

### A.1  Network Architecture

Figure 6 provides a more detailed depiction of our architecture, specifically focusing on the dimensions or sizes of individual layers.

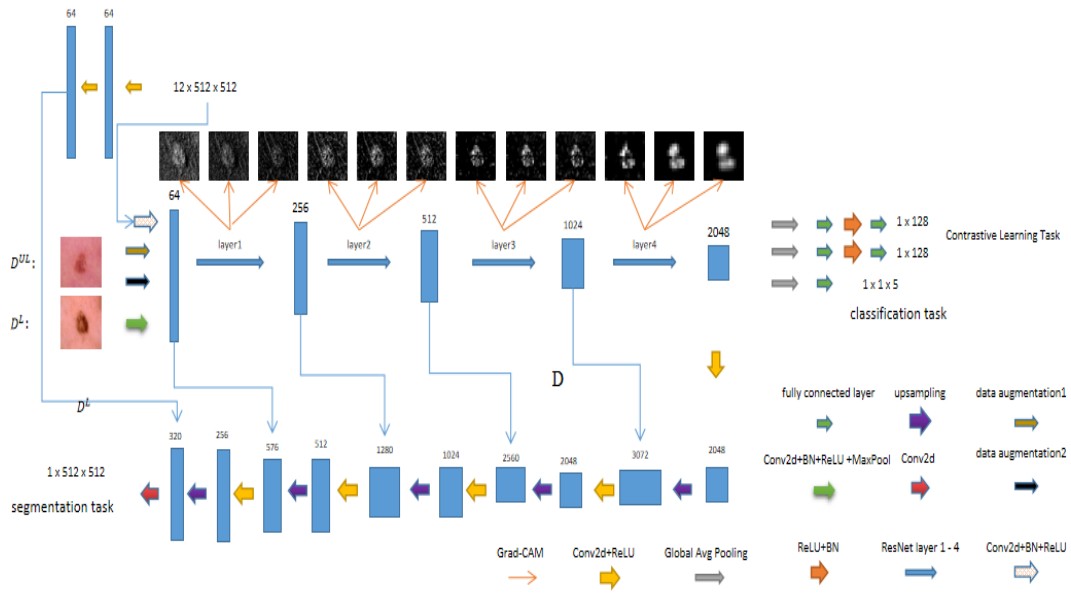

Figure 6: Proposed architecture for explainable diagnosis of melanoma: the architecture is trained to simultaneously classify skin lesion pictures using a CNN classifier and learn to localize melanoma clinical indicators on the input image using a U-Net-based segmentation network. The classification branch receive its input from the segmentation path to enforce classifying images based on clinical indicators.

### A.2  Hardware and Optimization Parameters

The complete framework has been instantiated through the utilization of the PyTorch library and underwent training on a system equipped with four NVIDIA RTX 2080Ti GPUs, each possessing 11GB of memory. Our chosen architecture, Bio-U-Net, incorporates ResNet50 as its encoder, leveraging pretrained weights from ImageNet to enhance its initial learning. Throughout the training phase, we employed the Adam optimizer with a consistent set of hyperparameters (learning rate = $1^{-4}$, weight decay = $1^{-4}$) across all tasks. This standardized optimization approach contributes to the stability and uniformity of the training process for various objectives within our framework.

|  | subnetwork(1)/SIMCLR | subnetwork(2)/Resnet | subnetwork(3)/Unet | Bio-U-Net/Total |
|---|---|---|---|---|
| FLOPs | 43.2 | 129.5 | 469.3 | 642.0 |

Table 7: Model Computational Complexity (in billion FLOPs).

The table above delineates the computational complexity of each subnetwork within the Bio-U-Net model, measured in billion FLOPs (Floating Point Operations per Second). Subnetwork 1, known as SIMCLR, requires 43.2 billion FLOPs and is generally associated with self-supervised learning in image recognition. Subnetwork 2, the Resnet, has a complexity of 129.5 billion FLOPs, a popular architecture in deep learning for image processing. Subnetwork 3, the Unet, with its 469.3 billion FLOPs, is integral for tasks like image segmentation, which is crucial in medical image analysis, such as identifying melanoma in skin lesion images. The total computational load of the Bio-U-Net model is 642.0 billion FLOPs. Despite this high computational requirement, the model's primary advantage lies in its one-time

training process. Once trained, it can efficiently process a variety of melanoma-related queries without the need for retraining. This feature not only saves significant computational resources and time but also leverages the individual strengths of each subnetwork for comprehensive and efficient melanoma analysis and diagnosis.

### A.3 Optimization Implementation Details

**Preprocessing:** The initial phase of our training process involves the training of the subnetwork $f_{CLR}(\cdot)$, wherein *Dataset B* acts as the input. To ensure uniformity, each image within *Dataset B* undergoes resizing to dimensions of $512 \times 512$. Subsequently, normalization is applied to achieve a zero mean and unit variance for the images. To enhance the model's generalization capabilities, we implement data augmentation techniques, encompassing random operations such as rotation, cropping, adjustments to brightness, contrast, saturation, and flipping.

Within each mini-batch, a positive example is paired with 23 negative examples, resulting in a batch size set at 24. This configuration aids in training the network with a more balanced representation of positive and negative instances. Additionally, the temperature parameter $\tau$ is fixed at a value of 0.5, contributing to the stability and effectiveness of the contrastive learning process during the training of $f_{CLR}(\cdot)$.

We then perform the classification task using all of *Dataset A*'s data, each resized to $512 \times 512$, and normalize images with zero mean and unit variance without performing any data augmentation. Due to the large size of the images and memory cap, the batch size is set to 16.

**Segmentation training:** Upon the successful conclusion of the classification task, our next step involves the utilization of Grad-CAM to generate an attention heatmap. This heatmap is created by inputting a bottleneck block along with the designated target attribute index into the Grad-CAM process. Subsequently, we systematically replace each bottleneck one at a time, repeating the process until all bottlenecks have been explored.

Following the acquisition of these heatmaps, our approach involves inputting them into the $f_{Seg}(\cdot)$ subnetwork, which is then trained to function as the reconstruction module. Throughout this training phase, a batch size of 8 is employed, ensuring efficiency and effectiveness in the learning process of the $f_{Seg}(\cdot)$ subnetwork. This iterative procedure, incorporating Grad-CAM and subsequent training, plays a crucial role in refining the overall performance and interpretability of our model.

Upon the conclusion of the network training phase, our subsequent step involves the generation of heatmaps through the application of Grad-CAM. Specifically, we denote these heatmaps as $f_{Grad_{C}AM}(b_i, j)$, where $b_i$ is an element of the set $E_1(\cdot)$. Here, $E_1(\cdot)$ represents the encoder, and $b_i$ refers to the bottleneck block. The encoder is loaded with the optimal checkpoint parameters obtained during training, and the index of the selected attribute is supplied as input.

To determine the final localization mask for the biomarker indicators, we specifically choose the heatmap associated with the last block in the process. This meticulous selection ensures that the generated heatmap accurately reflects the most relevant and discriminative features pertaining to the selected biomarker attributes. In essence, the Grad-CAM approach, coupled with the utilization of optimal checkpoint parameters, enhances the precision and interpretability of the final localization masks for biomarkers within our model.

## B  Additional Experimental Results

Illustrations from Figure 7 to Figure 10 provide additional instances of generated maps, enabling a more in-depth examination and comparison. The visual representations in these figures showcase that our methodology consistently produces maps that exhibit a notably higher degree of resemblance to those generated by dermatologists. This observation underscores the effectiveness and accuracy of our approach in generating maps that closely align with human-interpretable features.

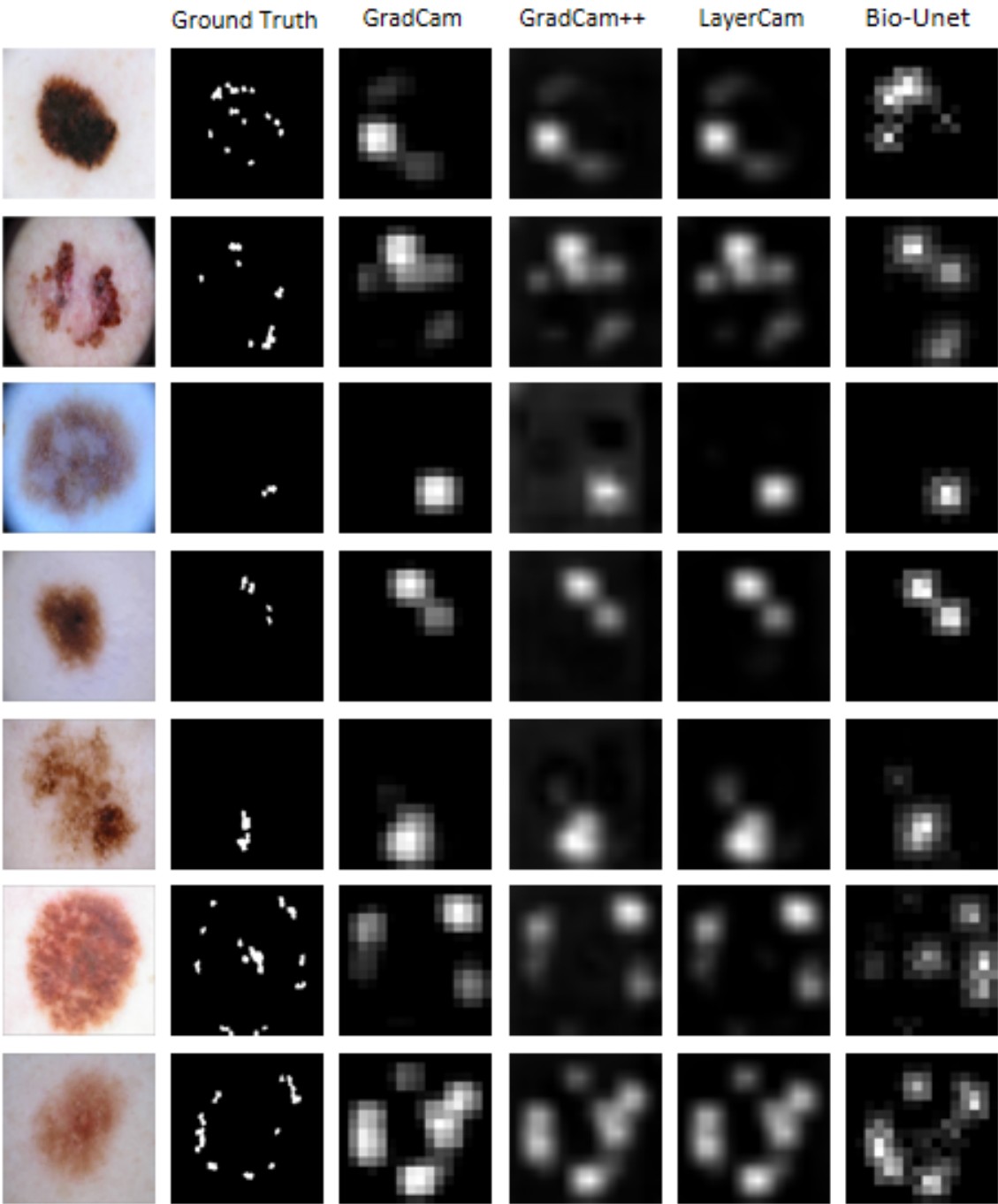

Figure 7: More examples of feature globules; Examples showing that $f_{Seg}$ can make the framework focus more on small important regions rather than large ones

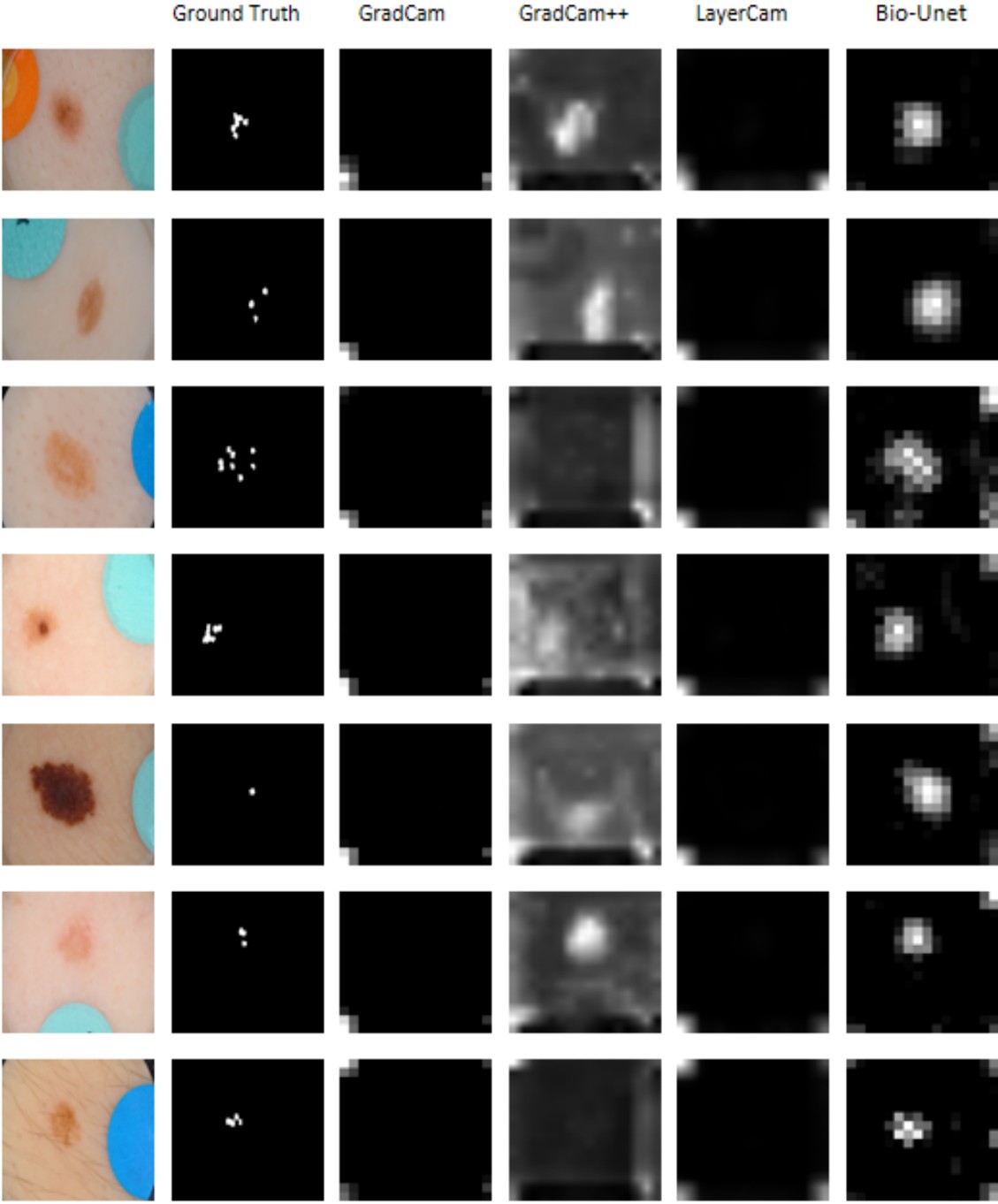

Figure 8: More examples of feature Milia like Cyst; Example showing that $f_{CLR}$ can help framewrok find large region of small scatter points and $f_{Seg}$ can make the framework focus more on small important regions rather than large ones

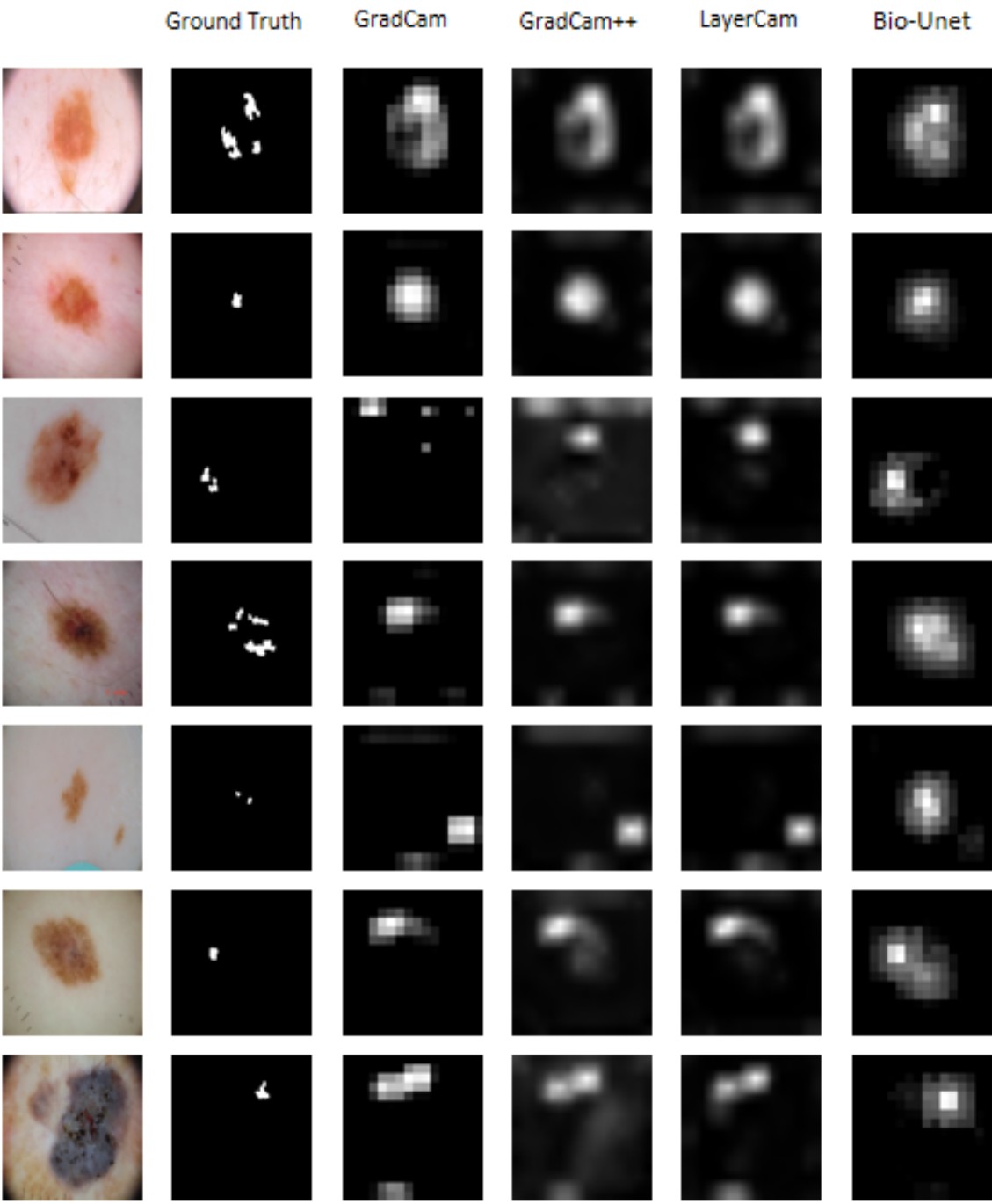

Figure 9: More examples of feature negative network; Example showing that $f_{Seg}$ can make the framework focus more on small important regions rather than large ones

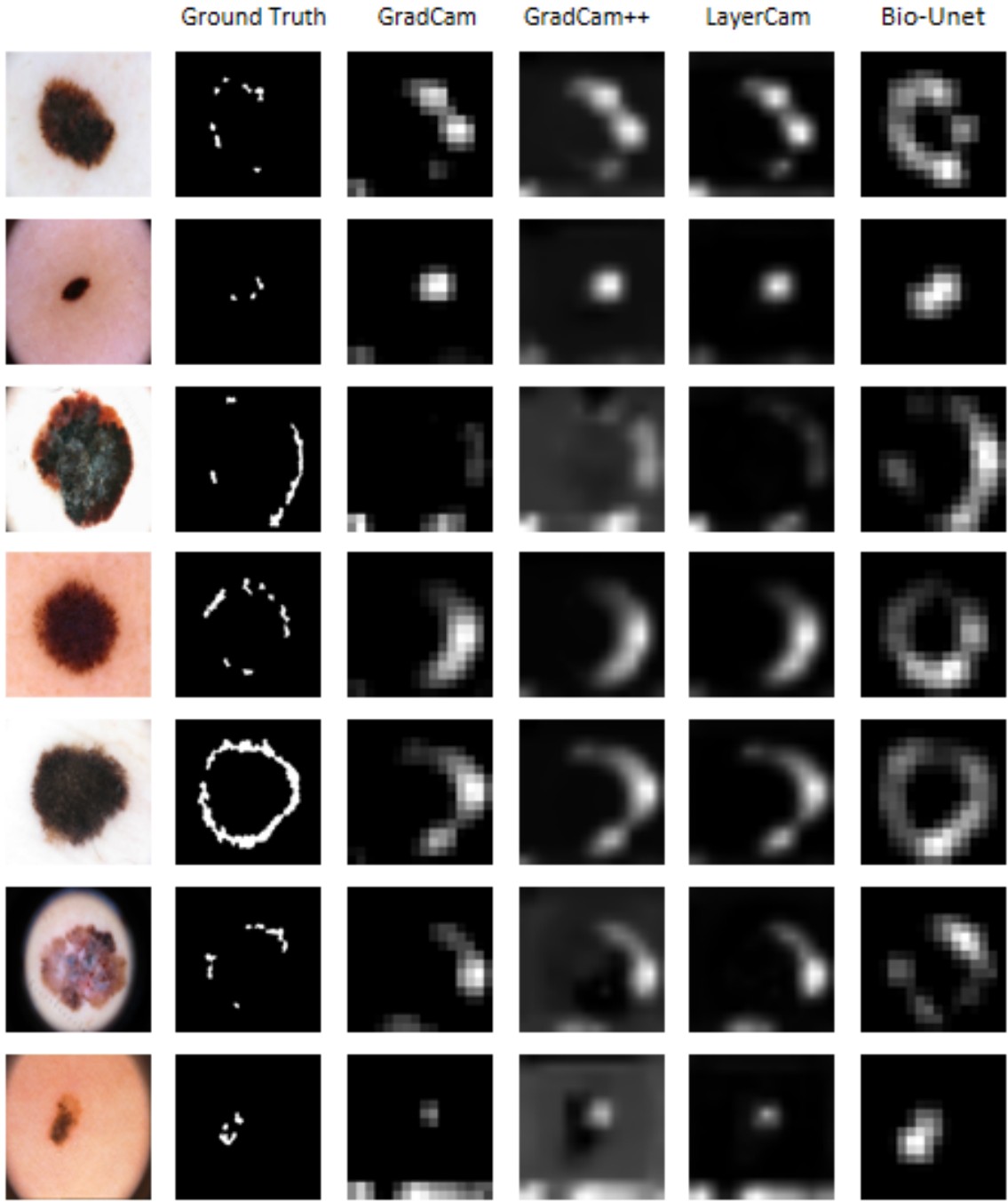

Figure 10: More examples of feature streaks; Example showing that $f_{CLR}$ can help framewrok find large region of small scatter points and $f_{Seg}$ can make the framework focus more on small important regions rather than large ones

