# OpenReview forum: "Explainable Diagnosis of Melanoma Based on  Localization of Clinical Indicators and Self-Supervised Learning"
_TMLR — Withdrawn by Authors_

### Review · Reviewer_PX7T · 2023-12-28

**Summary Of Contributions:**

The paper presents a DL architecture for melanoma diagnosis based on aligning saliency maps-based explanations with a given ground truth of the output to mimic expert dermatologists that reason on top of "features" instead of on top of pixels like traditional CNN approaches. The problem of having a few finely annotated data is overcome through self-supervised learning. The experiments show that the proposal works better than some other existing methods.

**Audience:**

Yes

**Claims And Evidence:**

No

**Requested Changes:**

The overview of the architecture 4.1 should be made more clear. There are discrepancies and/or different usage of names between the bullet list and the figure. Also, the anticipation of the solution in the last paragraph of Section 3 creates a bias in the dear that prevents a clear understanding of Section 4.1. Also, Fig 2 should report the same formal definitions in the description, the numbers (x) do not help in understanding and make misunderstandings. Indeed, I expected subnetwork (3) to be trained on Dataset A, but it does not seem so. For instance, the first sentence in 4.2 is very clear and should be presented earlier if the whole training is based on that.

In the experiments,, I expected a small validation with real users comparing explanations provided by the proposed approach against some standard XAI approaches.

Some related works are missing (and could be considered as competitors). In particular, I suggest a comparison against not only approaches based on saliency maps but also those based on counterfactual explanations.

Metta, C., Beretta, A., Guidotti, R., Yin, Y., Gallinari, P., Rinzivillo, S., & Giannotti, F. (2023). Improving trust and confidence in medical skin lesion diagnosis through explainable deep learning. International Journal of Data Science and Analytics, 1-13.

Dhurandhar, A., et al.: Explanations based on the missing: towards contrastive explanations with pertinent negatives, Advances in Neural Information Processing Systems, 592–603, (2018)

Bodria, F., Giannotti, F., Guidotti, R., Naretto, F., Pedreschi, D., & Rinzivillo, S. (2023). Benchmarking and survey of explanation methods for black box models. Data Mining and Knowledge Discovery, 1-60.

Minor issues.

In Table 2, it is not clear if the DICE metric should be high or low.

Another competitor for the proposal could be a traditional ML method (e.g. an SVM, Logistic Regressor or, Ensemble method) trained on the segmentations of the methods for the various features extracted. Indeed, what is basically doing the proposal, is a supervised feature extraction that help in identifying a certain class (that is the definition of Shapelets in time series).

**Strengths And Weaknesses:**

The paper is interesting and well-written overall. Some aspects reported below should be curated before acceptance.

---

> ### Author Response · Authors · 2024-01-16
> **Response to the reviewer**
>
> We thank the reviewer for identifying that our paper is interesting and well-written and hope that we can address the raised points adequately below. We have tried to address the raised concerns in the order written by the reviewer.
>
> 1. The overview of the architecture 4.1 should be made more clear...
>
> Response: Thank you for pointing out the need for clarity in the overview of Architecture 4.1. The usage of names has been made consistent between the bullet list and the figure. Furthermore, terminology from Section 3 has been incorporated into the overview of Architecture 4.1 for enhanced coherence.
>
>
> 2. Also, Fig 2 should report the same formal definitions in the description, the numbers (x) do not help in understanding and make misunderstandings.
>
> Response: Thank you for your feedback. The numbers (x) in Figure 2 have been replaced with 'ABCD', which are not used in other contexts, to avoid confusion. Additionally, the formal definitions have been integrated into Figure 2 for better clarity.
>
>
> 3. Indeed, I expected subnetwork (3) to be trained on Dataset A, but it does not seem so. For instance, the first sentence in 4.2 is very clear and should be presented earlier if the whole training is based on that.
>
> Response: Thank you for your suggestion. The sentence that clearly states the purpose of the paper has been moved to the beginning of Section 4. This change will help readers grasp the paper's objectives more clearly before proceeding with the rest of the section.
>
>
> 4. Raise point by the reviewer: In the experiments, I expected a small validation with real users ...
>
> Response: Comparing against real-user explanations can indeed improve the paper but unfortunately we are not dermatologists and finding dermatologists who are eager to collaborate with us in two weeks was not easy. But we would like to point out that the segmentation masks for the biomarkers that we use for validation are generated by expert dermatologists. Hence, when our algorithm is able to identify and localize the biomarkers when compared to maps generated by dermatologists, we present evidence that it is effective.
>
>
> 5. Raise point by the reviewer:  Some related works are missing ...
>
> Response: Thank you for pointing our attention to works that we have missed. We included these papers in our discussion but comparing our results against the mentioned papers is not feasible. The reason is that our work diverges in its core approach to explainable AI (XAI) in medical skin lesion diagnosis compared to the mentioned paper:
>
>  Metta et al. (2023): Unlike the work of Metta et al., which focused on entire lesion classification to enhance trust in medical diagnosis, our study specifically aims at attention heatmaps for each melanoma sub-attribute that are consistent with what experts believe. We achieve this by incorporating attribute masks during training, thus ensuring that the salience maps from the last convolutional layers closely resemble clinician-expected heatmaps. This methodology differs fundamentally from developing XAI tools like LIME or LORE. Our goal is to intrinsically train an explainable model whose interpretability aligns with expert medical understanding, rather than merely employing an explainability tool. In essence, while both studies contribute to the field of XAI, they address different aspects with distinct methodologies. Our work offers a unique perspective on developing inherently interpretable AI models in line with clinical expectations, which we believe is a significant contribution to the field.
>
> Dhurandhar et al. (2018): Our work has a distinct focus. We aim to align DNN models' attention mechanisms with clinical expertise, particularly by evaluating the framework through the last convolution layer. This approach inherently embeds interpretability to closely mirror medical decision-making. While Dhurandhar et al. contribute to explainable AI (XAI) through counterfactual explanations, our methodology centers on developing AI models whose interpretability aligns with clinician insights at a fundamental level. Hence, a direct comparison might not fully capture the unique aspects and objectives of our study.
>
> Bodria, F., Giannotti, F., Guidotti, R., Naretto, F., Pedreschi, D., & Rinzivillo, S. (2023). Benchmarking and survey of explanation methods for black box models. Data Mining and Knowledge Discovery, 1-60.
>
> Minor issues:
>
> 6. Raise point by the reviewer:  In Table 2, it is not clear if the DICE metric should be high or low.
>
> Response: We updated the caption of Figure 2.
>
> 7. Raise point by the reviewer:  Comparison with traditional ML method.
>
> Response: Unfortunately, traditional ML methods do not offer explainability and also need extensive feature engineering. As a result, using them to solve our problem is not possible. They are merely classification methods for which we need to have a good feature extracting method.

---

> ### Author Response · Authors · 2024-01-23
> **Request for continual discussion**
>
> Dear Reviewer,
>
> We reiterate our appreciation for your time. We think that your concerns can be addressed and respectfully ask you to read our response and if possible engage in discussion with us if you feel your concerns have not been addressed. Please let us know in case some of our responses are not convincing yet. We strongly believe that through continual discussion, there is a high chance that we can address your concerns. We are hopeful that your time allows continual discussion so you can make your final recommendation with a high confidence.
>
> Best,
> Our team

---

> ### Author Response · Authors · 2024-02-14
> **Follow-up Request**
>
> Dear Reviewer,
>
> We very much thank you for your time and effort. We do understand that you have a busy schedule but at the moment, it looks like that you opinion is very critical for our submissions. It looks like that our responses have convinced one of the reviewers but another reviewer is unconvinced. As a result, your opinion is going to be a tie-breaking vote in our submission. We respectfully ask you to read the rest of reviewers and our responses and finalize your decision accordingly. Of course, we think that we have addressed your concerns but please do engage in discussion if some of your concerns are not addressed yet but you think with more discussion we can address them. Thank you again for your time and efforts.
>
> Best,
>
> Our team

---

> > ### Comment · Reviewer_PX7T · 2024-02-15
> >
> > Dear all, I see your point, but at the current stage, the paper cannot be accepted, in my opinion, without integration of the various aspects detailed in the review. A rebuttal with Q&A cannot be sufficient to convince me to accept the paper, as various things are missing in the manuscript. I do believe that the next submission will be more successful.

---

> > > ### Author Response · Authors · 2024-02-15
> > > **Follow-up**
> > >
> > > Dear Reviewer,
> > >
> > > We thank you for your continual engagement. We certainly understand that the job of reviewers is evaluate submissions rigorously. We also think a rejection outcome is not necessarily a bad outcome. We have had past personal experiences in which, a rejection led to an improved version and later that manuscript was accepted in a much better shape. However, such an outcome necessitates a roadmap to prepare the next version. In the case of TMLR, resubmission is possible only if "a description of the changes made since." is provided. Hence, we will need to have concrete objectives to prepare our resubmission.
> > >
> > > We also think that although "a rebuttal with Q&A cannot be sufficient to convince" a reviewer but it can certainly help to come up with a roadmap to improve the paper for resubmission. In your case, could you please clarify:
> > >
> > > 1. What are "the various aspects detailed in the review" that we need to integrate? We think with the exception of the following two, we have addressed your comments:
> > >
> > > a. Comparison with "real users".  We have explained that coming up with clinical collaborators is not easy but is not impossible.
> > >
> > > b. Comparison against prior works by the reviewer. We explained why we cannot compare against these specific works in our response. Could please explain what type of comparison do you expect? Please be concrete and provide the experimental setup you expect.
> > >
> > > Given the above, if we can add human user validation and comparison with prior work according to a concrete expectation that you have, will we address your concerns. Please list any other expectations as well.
> > >
> > > Thank you,
> > >
> > > Our team

---

### Review · Reviewer_bUbw · 2023-12-31

**Summary Of Contributions:**

Skin lesions are warning sign for diagnosing melanoma at early stage, but which often led to  delayed diagnosis due to high similarities of cancerous and benign lesions at early stages of melanoma. The clinical adoption of Deep learning (DL) for this task has been quite limited, because DL models are often uninterpretable. This paper develops explainable DL architecture for melanoma diagnosis. The proposed architecture segments input images and generates clinically interpretable melanoma indicator masks that are then used for classification.

**Audience:**

Yes

**Claims And Evidence:**

Yes

**Requested Changes:**

Requested Changes
It would be better to provide more experiment details of hardware, software, and computational analysis.

**Strengths And Weaknesses:**

Strengths:
1. Explanability of DL is important and highly relates to TMLR.
2. The motivation "the size of the Dataset A may not be large enough for training segmentation models. The challenge in our formulation is that privacy concerns and high annotation costs of medical image datasets limits the size of publicly available instances of Dataset A that are well-annotated with indicator masks." is practical.
3. The new architecture design is well motivated.
4. The experimental analysis provides

Weaknesses:
1. Typos: Algorithm 1 is out of box.
2. The new model may have high computational complexity than baselines. The computation resource comsumption details are not provided.

---

> ### Author Response · Authors · 2024-01-16
> **Response to the reviewer**
>
> We thank the reviewer for identifying that our research is well-motivated and our experiments are convincing.
>
> 1. Typos: Algorithm 1 is out of box.
>
> Response: We corrected the algorithm format.
>
> 2. The new model may have high computational complexity than baselines. The computation resource comsumption details are not provided.
>
> Response: In order to fulfill the request by the reviewer, we reported the complexity of the proposed approach along with baselines in the appendix A.2. Please refer to that section in the updated manuscript and the discussion related to the new experiment. We hope the new experiments can address the raised concern.
>
> 3. Limitation to melanoma
>
> Response:  we respectfully ask the reviewer to check the literature and see there are numerous papers published in top-tier venues that address solely adopting AI to address melanoma, including, but not limited to the following very recent papers:
>
> 	A. Dermatologist-like explainable AI enhances trust and confidence in diagnosing melanoma, Nature Communications, 2023
>
> 	B. Towards Trustable Skin Cancer Diagnosis via Rewriting Model’s Decision, CVPR, 2023
>
> 	C. A pathology deep learning system capable of triage of melanoma specimens utilizing dermatopathologist consensus as ground truth, ICCV, 2021
>
> 	D. Using Whole Slide Image Representations from Self-supervised Contrastive Learning for Melanoma Concordance Regression, ECCV, 2022
>
> We hope that the reviewer considers the above papers and compares them along with our work to see that our contribution is comparable. A quick search also will indicate there are far more similar papers in the literature since 2017. While extending our work to other types of diseases may be feasible, we hope that the reviewer considers the above work and conclude that addressing melanoma

---

> > ### Comment · Reviewer_bUbw · 2024-02-13
> > **Thanks for responses**
> >
> > Thanks for responses. My previous questions and concerns are addressed.

---

> > > ### Author Response · Authors · 2024-02-14
> > > **Thank you note**
> > >
> > > Dear Reviewer,
> > >
> > > Thank you for continual engagement. We are glad that we were able to address your concerns. We would very much appreciate if if you could highlight our strength in discussions with other reviewers.
> > >
> > > Best,
> > >
> > > Our team

---

> ### Author Response · Authors · 2024-01-23
> **Request for continuing discussion**
>
> Dear Reviewer,
>
> We reiterate our appreciation for your time. We think that your concerns can be addressed and respectfully ask you to read our response and if possible engage in discussion with us if you feel your concerns have not been addressed. Specifically, we hope you check the literature and see that there are tens of more published papers in respected conferences and journals, in addition to those listed in our response, that focus on adopting AI for diagnosing "melanoma" which demonstrates that only addressing melanoma in our work is a decent contribution. We are hopeful that your time allows continual discussion so you can make your final recommendation with a high confidence.
>
> Best,
> Our team

---

### Review · Reviewer_qXB6 · 2024-01-01

**Summary Of Contributions:**

The paper presents an explainable deep learning (DL) architecture for melanoma diagnosis, employing self-supervised learning to generate clinically interpretable melanoma indicator masks for classification. This approach addresses the challenge of data annotations and aims to provide more transparent decision-making in clinical settings.

**Audience:**

Yes

**Claims And Evidence:**

Yes

**Requested Changes:**

See Weaknesses

**Strengths And Weaknesses:**

### Strengths
- Comprehensive Framework: The Bio-U-Net architecture is comprehensive, particularly in its consideration of explainability. This aspect is crucial as it enhances the transparency and trustworthiness of the model in clinical applications.
- Effective Results: The framework demonstrates effective performance. This is significant, as achieving high accuracy in medical diagnosis models directly impacts their practical usefulness and reliability.

### Weaknesses
- Limited Innovation: The methodology primarily employs existing techniques, lacking significant innovation. In a field that is rapidly evolving, such as deep learning for medical diagnosis, more novel approaches or significant improvements over existing methods are generally expected to constitute a substantial contribution.
- Explainability of Results, Not Model: The paper achieves explainability at the level of results but does not extend this to the model itself. Model-level explainability is arguably more meaningful as it can provide users with a deeper understanding and thus greater confidence in the model. This aspect is especially important in medical applications where trust and clarity are paramount.
- Focus Restricted to Melanoma: The research is limited to melanoma diagnosis and does not consider a broader range of skin diseases. In clinical practice, a more comprehensive diagnostic tool capable of addressing a variety of skin conditions would be more desirable and useful.

---

> ### Author Response · Authors · 2024-01-16
> **Response to the reviewer**
>
> Thank you for recognizing that our results are effective and that our approach is comprehensive. We hope that we can address your concerns.
>
> 1. Raise point by the reviewer: Limited Innovation: The methodology primarily employs existing techniques, lacking significant innovation. In a field that is rapidly evolving, such as deep learning for medical diagnosis, more novel approaches or significant improvements over existing methods are generally expected to constitute a substantial contribution.
>
> Response: We agree that we are using existing techniques but please note that our novelty lies in developing a new architecture that is able to follow dermatologists for diagnosing melanoma. In this our work is novel. We respectfully ask the reviewer to check the literature and see that our work is novel in this sense. We also would like to mention that the comment by the reviewer is very broad and it is not easy to come up with a more novel approach in two weeks.
>
>
> 2. Raise point by the reviewer:  Explainability of Results, Not Model: The paper achieves explainability at the level of results but does not extend this to the model itself. Model-level explainability is arguably more meaningful as it can provide users with a deeper understanding and thus greater confidence in the model. This aspect is especially important in medical applications where trust and clarity are paramount.
>
> Response: We agree with the reviewer that model explainability is important but explainability of the results also is crucial in adopting AI in medical applications. In other words, our novelty is to propose a method for diagnosing melanoma such that the results are explainable. For a physician, this aspect is extremely important even in the absence of model explainability.
>
>
> 3. Raise point by the reviewer: Focus Restricted to Melanoma: The research is limited to melanoma diagnosis and does not consider a broader range of skin diseases. In clinical practice, a more comprehensive diagnostic tool capable of addressing a variety of skin conditions would be more desirable and useful.
>
> Response: Unfortunately, two weeks is too short to expand our results beyond melanoma. Please note that different skin conditions have different diagnosis protocols. We respectfully ask the reviewer to check the literature and see that there are many existing works that us AI solely for melanoma diagnosis, including, but not limited to the following very recent papers:
>
>
> 	A. Dermatologist-like explainable AI enhances trust and confidence in diagnosing melanoma, Nature Communications, 2023
>
> 	B. Towards Trustable Skin Cancer Diagnosis via Rewriting Model’s Decision, CVPR, 2023
>
> 	C. A pathology deep learning system capable of triage of melanoma specimens utilizing dermatopathologist consensus as ground truth, ICCV, 2021
>
> 	D. Using Whole Slide Image Representations from Self-supervised Contrastive Learning for Melanoma Concordance Regression, ECCV, 2022
>
> We hope that the reviewer considers the above papers and compares them along with our work to see that our contribution is comparable. A quick search also will indicate there are far more similar papers in the literature since 2017. While extending our work to other types of diseases may be feasible, we hope that the reviewer considers the above work and conclude that addressing melanoma

---

> ### Author Response · Authors · 2024-01-23
> **Request for continuing discussion**
>
> Dear Reviewer,
>
> We reiterate our appreciation for your time. We think that your concerns can be addressed and respectfully ask you to read our response and if possible engage in discussion with us if you feel your concerns have not been addressed. Specifically, we hope you check the literature and see that there are tens of more published papers in respected conferences and journals, in addition to those listed in our response, that focus on adopting AI for diagnosing "melanoma" which demonstrates that only addressing melanoma in our work is a decent contribution. We are hopeful that your time allows continual discussion so you can make your final recommendation with a high confidence.
>
> Best,
> Our team

---

### Author Response · Authors · 2024-01-16
**Note to the reviewers**

Dear Reviewers,

We are thankful for your time and effort. We understand that reviewers spend their time on a fully volunteer basis and we appreciate your reviews which helped improving our work. We have provided our response to each reviewer. We respectfully ask the reviewers to read our responses, engage in discussions in this stage, and give us a chance to improve our work and hopefully raise our work to the acceptance level with your guidance.

Best,

Our team

---

### Note · Authors · 2024-02-28

I have read and agree with the venue's withdrawal policy on behalf of myself and my co-authors.